# Comparison of Minimally Invasive versus Open Pancreatoduodenectomy for Pancreatic Ductal Adenocarcinoma: A Propensity Score Matching Analysis

**DOI:** 10.3390/cancers12040982

**Published:** 2020-04-15

**Authors:** Jaewoo Kwon, Ki Byung Song, Seo Young Park, Dakyum Shin, Sarang Hong, Yejong Park, Woohyung Lee, Jae Hoon Lee, Dae Wook Hwang, Song Cheol Kim

**Affiliations:** 1Division of Hepato-Biliary and Pancreatic Surgery, Department of Surgery, University of Ulsan College of Medicine and Asan Medical Center, 88, Olympic-Ro 43-Gil, Songpa-Gu, Seoul 05505, Korea; skunlvup@naver.com (J.K.); mtsong21c@naver.com (K.B.S.); gracedkshin@gmail.com (D.S.); 8thofnovember@hanmail.net (S.H.); blackpig856@gmail.com (Y.P.); ywhnet@gmail.com (W.L.); gooddr23@naver.com (J.H.L.); drdwhwang@gmail.com (D.W.H.); 2Department of Clinical Epidemiology and Biostatistics, Asan Medical Center, University of Ulsan College of Medicine, 88, Olympic-Ro 43-Gil, Songpa-Gu, Seoul 05505, Korea; biostat81@gmail.com

**Keywords:** pancreatic cancer, pancreatoduodenectomy, minimally invasive surgery

## Abstract

*Background:* Few studies have compared perioperative and oncological outcomes between minimally invasive pancreatoduodenectomy (MIPD) and open pancreatoduodenectomy (OPD) for pancreatic ductal adenocarcinoma (PDAC). *Methods*: A retrospective review of patients undergoing MIPD and OPD for PDAC from January 2011 to December 2017 was performed. Perioperative, oncological, and survival outcomes were analyzed before and after propensity score matching (PSM). *Results*: Data from 1048 patients were evaluated (76 MIPD, 972 OPD). After PSM, 73 patients undergoing MIPD were matched with 219 patients undergoing OPD. Operation times were longer for MIPD than OPD (392 vs. 327 min, *p* < 0.001). Postoperative hospital stays were shorter for MIPD patients than OPD patients (12.4 vs. 14.2 days, *p* = 0.040). The rate of overall complications and postoperative pancreatic fistula did not differ between the two groups. Adjuvant treatment rates were higher following MIPD (80.8% vs. 59.8%, *p* = 0.002). With the exception of perineural invasion, no differences were seen between the two groups in pathological outcomes. The median overall survival and disease-free survival rates did not differ between the groups. *Conclusions*: MIPD showed shorter postoperative hospital stays and comparable perioperative and oncological outcomes to OPD for selected PDAC patients. Future randomized studies will be required to validate these findings.

## 1. Introduction

Minimally invasive surgery has become the standard of care for many surgical procedures across different specialties and is currently standard procedure for the resection of intraabdominal organs, including the stomach [1,2], gallbladder [3], spleen [4,5], colon [6,7], and kidney [8,9]. However, its use in pancreatic surgery has been limited because of the complexity of these operations. Recently, minimally invasive surgery for benign and malignant pancreatic tumors has gained wider acceptance and is attracting more research attention [10,11,12]. In addition, a Pan-European propensity score matched study was published that showed a comparable survival outcome when performing minimal invasive distal pancreatectomy for pancreatic cancer [13]. However, minimally invasive pancreatoduodenectomy (MIPD), which includes laparoscopic pancreatoduodenectomy (LPD) and robotic pancreatoduodenectomy (RPD), remains limited by a lack of generalizability, and open surgery is preferred for pancreatic ductal adenocarcinoma (PDAC) due to concerns about adequate oncological outcomes and the potential for vessel resection. Recent studies have investigated LPD for PDAC [12,14]. Until now, however, no matched analyses have been conducted to analyze the perioperative and long-term oncologic outcomes of MIPD versus open pancreatoduodenectomy (OPD) for PDAC. The present retrospective study has compared perioperative and oncological outcomes between MIPD and OPD for PDAC using propensity score matching (PSM) analysis. 

## 2. Materials and Methods

### 2.1. Patients and Post-Surgical Monitoring

Data from patients treated by pancreatoduodenectomy (MIPD or OPD) for PDAC between January 2011 and December 2017 at the Asan Medical Center were evaluated. MIPD was defined in this study as LPD or RPD. Patients who presented with benign lesions or periampullary malignancy other than PDAC were excluded from the study, as were patients presenting with other types of pancreatic cancer (adenosquamous carcinoma, colloid carcinoma, hepatoid carcinoma, medullary carcinoma, signet ring cell carcinoma, undifferentiated carcinoma, and undifferentiated carcinoma with osteoclast-like giant cells). Patient selection is summarized in Figure 1. 

Data on eligible patients were obtained from electronic medical records and reviewed retrospectively. The following clinicopathological data were collected: age, sex, body mass index (BMI), American Society of Anesthesiologists (ASA) physical status classification, carbohydrate antigen 19-9 (CA19-9), carcinoembryonic antigen (CEA), preoperative biliary drainage, modified Glasgow Prognostic Score (mGPS) [15], surgical procedure, concurrent vessel resection, concurrent resection of other organs, neoadjuvant chemotherapy, year of surgery, operation time, postoperative complications, postoperative pancreatic fistula (POPF), delayed gastric emptying, post-pancreatectomy hemorrhage, biliary stricture, reoperation history, 90 day mortality, postoperative hospital stay, adjuvant chemotherapy, interval between adjuvant treatment and surgery, pathological findings, tumor size, TNM stage (American Joint Committee on Cancer Stage, 8th edition), overall survival, and disease-free survival. The resection margin status was categorized as R0 or R1. If the closest safe resection margin was < 1 mm, it was categorized as R1 [16]. Postoperative complications were classified according to the Clavien-Dindo system [17]. Late complications were defined as complications that were found in outpatients after discharge. POPF was graded according to the definition of the International Study Group of Pancreatic Surgery (ISGPS), which was updated in 2016 [18]. Delayed gastric emptying and post-pancreatectomy hemorrhage were diagnosed in accordance with ISGPS parameters [19,20]. Biliary stricture was defined as postoperative jaundice resulting in treatment with percutaneous biliary drainage, balloon cholangioplasty, and stent insertion. Postoperative surveillance with contrast-enhanced abdominoperineal computed tomography (CT) evaluation and CA19-9 level tests was conducted every 3 months for the first 2 years following surgery, and then every 6 months thereafter in all patients. Recurrence was diagnosed based on detection of new progressive lesions on abdominal CT and an increase in CA19-9 levels. When lesions signifying potential recurrent disease were detected, ^18^F-fluorodeoxyglucose positron emission tomography (FDG-PET) and/or chest CT were performed along with biopsy to confirm the diagnosis if a differential diagnosis was required. Overall survival (OS) was measured from the time of surgery until death or the date of national insurance loss. The last follow-up was performed in March 2019. The retrospective cohort study design was approved by the institutional review board of the Asan Medical Center (approval number: 2019-0683).

### 2.2. Surgical Indications for MIPD in Patients with PDAC 

From 2007 to 2017, 552 cases of MIPD for benign or malignant lesion were performed in our center by a total of five surgeons, each of whom performed at least 70 cases of pancreaticoduodenectomy per year [21]. MIPD for PDAC has been conducted since 2011, after an accumulation of laparoscopic pancreatoduodenectomies for benign lesions that began in 2007 [21]. Patients diagnosed with resectable PDAC with preserved fat planes between the tumor and celiac axis, hepatic artery, and superior mesenteric artery, and without previous abdominal surgery were eligible for MIPD. If the lesion changed from locally advanced to resectable PDAC after neoadjuvant chemotherapy, the patient was also scheduled for MIPD. Prior consent for MIPD was obtained from all eligible patients. 

### 2.3. Surgical Technique

#### 2.3.1. Laparoscopic Pancreatoduodenectomy 

The patient was placed in a supine position, and an anti-Trendelenburg (10–30°) was used to expose the surgical field. Two monitors were placed on both sides of the patient. The surgeon and laparoscopist stood to the right of the patient, with the assistant positioned to the left. The surgeon’s right-hand port (12 mm) was inserted through the left side of the umbilicus; an additional four trocars were then placed (Figure 2). 

After abdominal access was established, the greater omentum was divided using an energy device, and the right colon was separated and fully mobilized from the liver and duodenum. The retropancreatic superior mesenteric vein (SMV) was then exposed, and the right gastroepiploic vessels were transected. After removing soft tissue from around the SMV and superior mesenteric artery, each was hung with a vessel loop. The mobilization of the duodenum to the Treitz ligament was performed with traction of the duodenum by the surgical assistant. The stomach and duodenum were divided using an endoscopic linear stapler.

After cholecystectomy, dissection of the hepatoduodenal ligament and isolation of the common bile duct were performed. The right and left hepatic arteries were identified and isolated, and lymph node dissection was performed. The gastrohepatic ligament was opened to visualize the superior border of the pancreas and to identify the common hepatic artery. The right gastric artery and gastroduodenal artery were identified and transected using a Hem-o-lock clip. The pancreas was divided above the SMV using an energy device. After retracting the resected pancreas to the right side of the patient’s abdomen, the portal vein was identified and hung with a vessel loop. The jejunum was divided approximately 10–15 cm distal to the Treitz ligament with an endoscopic linear stapler. An energy device and endoscopic electrocautery were used to separate the remaining soft tissue and branches from the superior mesenteric artery between the uncinate process of the pancreas and the superior mesenteric artery to complete the resection. Pancreatojejunostomy was performed using a double-layered, end-to-side, duct-to-mucosa method by laparoscopic suture. A polyethylene internal stent was inserted into the pancreatic duct. End-to-side hepaticojejunostomy was performed using laparoscopic continuous suturing at the posterior wall and interrupted or continuous suturing at the anterior wall. Duodenojejunostomy or gastrojejunostomy with jejunojejunostomy was performed intracorporeally or extracorporeally via the specimen extraction site. Two or three closed suction drains were placed at the superior and inferior borders of the pancreatojejunostomy site.

#### 2.3.2. Robotic Pancreatoduodenectomy 

The patient was positioned as described for LPD. RPD was undertaken using one of two major surgical procedures. The first was LPD with robotic reconstruction, which was performed as per LPD to resect tumors. The surgeon’s positions and trocar locations (Figure 2A) were the same as for LPD. After the resection phase, the duct-to-mucosa (or dunking) pancreatojejunostomy and end-to-side hepaticojejunostomy reconstructions were performed using a robotic system. The surgeon’s two ports and the assistant port were replaced with a robotic 8 mm port that inserted into the robotic arm, followed by completely robotic pancreatoduodenectomy. 

The second type of robotic pancreatoduodenectomy was resection and anastomosis, performed by robotic system. Four robotic trocars, including a 12 mm camera port and two accessory laparoscopic ports for the assistant, were used during the operation (Figure 2B). The surgical procedure was the same as that in LPD, except that the robot arm was used. After hepaticojejunostomy, the specimen was extracted through the extended robot camera port site. Duodenojejunostomy or gastrojejunostomy with jejunojejunostomy was performed extracorporeally, in the same way as LPD.

### 2.4. Statistical Analysis 

Demographics, perioperative, pathological, and oncological outcomes were compared between the MIPD and OPD groups. Continuous variables were reported as the mean and standard deviation, or median and range as appropriate, and were compared using Student’s *t*-test. Categorical variables were compared using the chi-square test, Fisher’s exact test, or linear-by-linear association test. All tests were two-sided and a *p*-value ≤ 0.05 was considered statistically significant. Survival curves were generated using the Kaplan–Meier method. Comparison of survival between the MIPD and OPD groups was performed with the log rank test. A PSM analysis was performed, wherein 73 patients undergoing MIPD were matched with 219 patients undergoing OPD to mitigate the limitations of a retrospective study and selection bias. PSM analysis reduces the impact of treatment-selection bias on the estimation of causal treatment effects in a retrospective cohort study. Propensity scores were estimated by fitting a logistic regression model with the OP type (MIPD vs. OPD) as the response variable. Two continuous variables including age and BMI, and ten categorical variables including sex, ASA score (grade I–III), CA19-9 range (normal or increased [>37 U/mL]), CEA range (normal or increased [>5 ng/mL]), preoperative drainage (yes or no), mGPS (0, 1, or 2), neoadjuvant chemotherapy (yes or no), concurrent vessel resection (yes or no), concurrent other organ resection (yes or no), and operation year (before or after 2015) were included as independent variables. Multiple imputations with m = 5 were performed, and the propensity score model was fitted separately for each of the five imputed (thus complete) datasets to account for gaps in the data for some of the independent variables. For each patient, the average of the five estimated propensity scores was calculated, and this averaged score was used for matching. This matching was performed at a ratio of 1:3 MIPD to OPD, using a width of 0.2 standard deviations of the logit of the estimated propensity score. After PSM, there was a difference in the pathological result, which also impacts on survival outcome, so PSM including pathological results was performed once again to confirm more clearly whether survival could be affected by the surgical method, even though this result is not statistically correct. Kaplan–Meier survival curves were analyzed for OS and disease-free survival (DFS) rates. A robust estimator was used to allow for clustering effects within matched stratum for inference based on Cox regression. The prognostic effects of MIPD were estimated using a multivariate Cox proportional hazard model. Statistical analyses were calculated and compared using SPSS 21.0 (IBM Corp., Armonk, NY, USA) and R 3.5.1 (R Foundation for Statistical Computing, Vienna, Austria). 

## 3. Results

### 3.1. Patient Demographics Prior to PSM

Between January 2011 and December 2017, a total of 3059 patients underwent pancreatoduodenectomy. Following exclusions, 1048 patients (652 male, 396 female) were enrolled in the study. Table 1 shows the demographics of the study cohort grouped by MIPD (*n* = 76, including 11 cases of RPD) and OPD (*n* = 972). Age, sex, BMI, ASA score, proportion of elevated CEA, proportion of mGPS, neoadjuvant chemotherapy, and concurrent resection of other organs did not differ significantly between the MIPD and OPD groups. Increased levels of CA19-9 were seen in 50.0% of the MIPD group and 65.7% of the OPD group (*p* = 0.017). The number of patients undergoing preoperative biliary drainage and concurrent vessel resection was higher in the OPD group than in the MIPD group (47.4% versus 62.0%, respectively; *p* = 0.016 and 15.8% versus 36.9%, respectively; *p* < 0.001). Among the 12 cases of concurrent vessel resection in the MIPD group, eight were portal vein or SMV wedge resection with primary closure, two were end-to-end resection with synthetic graft anastomosis, one was portal vein wedge resection with patching, and one case was right hepatic artery resection with non-anastomosis. Since 2015, 86.8% of patients underwent MIPD and 46.5% underwent OPD (*p* < 0.001).

### 3.2. Comparison of Perioperative and Oncological Outcomes in the MIPD and OPD Groups Prior to PSM

Table 2 shows the perioperative outcomes of pancreatoduodenectomy in both groups. None of the patients who received MIPD had extracorporeal pancreaticojejunostomy or hepaticojejunostomy; one case required open conversion because of portal vein and right hepatic artery invasion. The mean follow-up period was 26.79 months (median, 20.47; range, 0.13–97.61 months). Operation time for MIPD and OPD showed a statistically significant difference (392 vs. 368 min, respectively; *p* = 0.043). Although overall complication rates and in-hospital complications showed no differences between the two groups, late complications were more frequent in the MIPD group (9.2% vs. 4.9% in the OPD group; *p* = 0.021). Clinically relevant POPF, delayed gastric emptying grade B or C, and post-pancreatectomy hemorrhage grade B or C did not vary significantly between the groups, but biliary stricture occurred more frequently after MIPD than OPD (5.3% vs. 0.7%, respectively; *p* = 0.006). No 90-day mortalities were observed in the MIPD group, but seven occurred in the OPD group, although this difference was not statistically significant. Four patients died as a result of disease progression with multiple metastases; three were discharged without complications, but death was confirmed through loss of national insurance. The postoperative hospital stay was shorter for MIPD patients than OPD patients (12.2 vs. 15.0 days, respectively; *p* < 0.001). More MIPD patients than OPD patients received adjuvant treatments (80.3% vs. 68.1%, respectively; *p* = 0.001). The method of adjuvant chemotherapy regimen was similar between the two groups, but the proportion of gemcitabine-based regimens was higher in MIPD patients, and fluoropyrimidine was more common in OPD. The period from surgery to adjuvant treatment did not differ between the two groups.

Table 3 shows the pathological outcomes following MIPD and OPD. Tumor sizes were smaller in the MIPD group (2.7 vs. 3.1 cm in the OPD group; *p* = 0.019). The rate of perineural invasion was higher in the OPD group than in the MIPD group (69.7% vs. 87.7%, respectively; *p* < 0.001). The number of harvested lymph nodes and positive lymph nodes were larger in the OPD than in the MIPD group (18.6 vs. 22.1, respectively; *p* = 0.006 and 1.5 vs. 2.0, respectively; *p* = 0.041); however, the positive lymph node ratios did not differ between the groups. No other factors showed a significant difference between the MIPD and OPD groups. 

Figure 3 shows Kaplan–Meier survival curves for the MIPD and OPD groups. The median OS and DFS rates showed no significant differences between the two groups. 

### 3.3. Comparative Analysis of Perioperative and Oncologic Outcomes in the MIPD and OPD Groups After PSM 

Table 4 shows matched demographic data for the two groups; all variables were well matched. 

Table 5 shows comparative data for perioperative outcomes between the two groups after matching. Operation times were shorter for OPD than MIPD (392 vs. 327 min, respectively; *p* < 0.001). As seen, prior to PSM matching, overall complication rates were similar, and the proportion of in-hospital and late complications did not differ between the two groups. The incidence of clinically relevant POPF, delayed gastric emptying grade B or C, and post-pancreatectomy hemorrhage grade B or C did not vary significantly between the matched groups. Biliary stricture and reoperation also showed no differences after PSM. Postoperative hospital stay following MIPD was shorter than following OPD (12.4 vs. 14.2 days, respectively; *p* < 0.040). More MIPD patients than OPD patients received adjuvant treatment, (80.8% vs. 59.8%, respectively; *p* = 0.002). The adjuvant regimen and the period from surgery to adjuvant treatment showed no differences between the two groups.

Table 6 summarizes the pathological results after PSM. Tumor sizes were similar between the two groups, and TNM stage, tumor differentiation, and lymphovascular invasion did not vary significantly. The number of harvested lymph nodes was smaller in MIPD group, but the difference was not statistically significant (18.9 vs. 21.3, respectively; *p* = 0.073). The rate of perineural invasion was higher in the OPD group than in the MIPD group (81.7% vs. 69.9%, respectively; *p* = 0.042).

The median OS and DFS were similar in the two groups (Figure 4). Estimated 1-, 2- year OS rate was 84.9% (95% CI, 77.1%–93.5%), 59.8% (95% CI, 48.3%–74.1%) in MIPD group and 79.4% (95% CI, 74.2%–84.9%), 54.0% (95% CI, 47.4%–61.6%) in OPD group. There was no difference in survival rate between two groups (*p* = 0.143). 

Estimated 1-, 2- year DFS rate was 55.7% (95% CI, 45.4%–68.5%), 33.8% (95% CI, 23.8%–48.1%) in MIPD groups and 45.5%(95% CI, 39.1%–53.0%), 31.3% (95% CI, 23.8%–48.1%) in OPD group. No notable differences were seen in DFS between two groups (*p* = 0.278).

### 3.4. Multivariable Model of Prognostic Factors for OS and DFS After PSM

Differences in pathological results after PSM and survival risk factors were assessed using a multivariate Cox proportional hazards model to confirm whether MIPD affected OS (Table 7) and DFS (Table 8) in 73 MIPD patients and 219 OPD patients after PSM. This analysis showed that MIPD did not affect OS (95% CI, 0.525–1.175; *p* = 0.240) or DFS (95% CI, 0.523–1.150; *p* = 0.206). Differentiation, perineural invasion and resection margin status were prognostic factors for OS, and N1 stage and differentiation were prognostic factors for DFS.

### 3.5. Comparative Analysis of Oncologic Outcomes Between Resectable MIPD and OPD Groups after PSM for Pathologic Outcome

OS and DFS were also compared between the MIPD and OPD groups after PSM for pathological outcomes because of differences in perineural invasion. In this analysis, 66 MIPD patients and 132 OPD patients were enrolled for PSM. Pathologic outcome including perineural invasion were well matched after pathological PSM (Table 9). There were no statistically significant differences in OS and DFS between the two groups (Figure 5).

## 4. Discussion

The development of MIPD was based on Gagner and Pomp’s original LPD description from 1994 [22]. Several studies from high-volume centers have reported that MIPD might be feasible and confer advantages over OPD for benign lesions and periampullary malignancy [23,24,25]. However, reservations concerning the safety of MIPD persist as the majority of hospitals performing MIPD were low-volume centers, and its use has been associated with increased morbidity and mortality [26,27,28]. In addition, MIPD for PDAC is yet to show generalizable indication because of the complex relationships between major surrounding structures, inflammatory changes around the head of the pancreas, and invasion of major vessels. Stauffer et al. reported that 24.1% of LPD cases were converted to OPD after vein resection or adherence to the underlying vasculature resulting from desmoplastic or pancreatitis reactions [14]. In the current study, the OPD group presented with a higher mean level of CA19-9 and higher rates of preoperative drainage and concurrent vessel resection than the MIPD group. Differences in tumor size, T stage, proportion of perineural invasion, and number of positive lymph nodes also suggest selective indications for MIPD for PDAC. These findings suggest that MIPD is being performed in patients with less inflammation and less aggressive tumors than those undergoing OPD. Although our center has gradually expanded the indications of MIPD for PDAC, OPD remains the standard for pancreatoduodenectomy, with MIPD selection based on limited medical indications and surgeon preference. The current study therefore matched preoperative findings using PSM, and subsequently compared perioperative and oncologic outcomes between the MIPD and OPD groups.

Perioperative outcomes showed that there were differences in the operation time in the non-corrected data and PSM data. This result suggests that regardless of disease severity, MIPD is a more lengthy procedure because of technical difficulties; this result is also similar to the findings of previous reports [14,23,29,30]. However, despite the longer operation time of MIPD, overall complication and in-hospital / late complication rates did not differ between the two groups. Stauffer et al. have previously reported that the rate of postoperative complications did not differ between LDP and OPD for PDAC [14]. This result was also similar in a previous randomized clinical trial of LPD for periampullary disease [31,32].

In the current study, the duration of postoperative hospitalization was shorter after MIPD than OPD. Several reports have suggested that LPD for PDAC results in faster recovery times and an earlier return to activity than OPD [12,14]. Two randomized clinical trials of LPD also reported that laparoscopy offered a shorter hospital stay [31,32]. This can be explained by the lower levels of postoperative inflammation after laparoscopic surgery [33]. Other studies have reported that minimally invasive surgery minimized surgical stress in a cohort of patients where the absence or reduction of postoperative pain was essential for postoperative mobilization [34,35]. Randomized controlled trials exploring laparoscopic surgery for colon cancer have reported a positive impact on postoperative restitution with earlier recovery of bowel function [36,37]. Less postoperative inflammation, a reduction in postoperative pain, and increased mobilization after minimally invasive surgery could explain the shorter hospital stays observed in the MIPD group of the current study.

Adjuvant chemotherapy is a critical component of treatment for patients with PDAC. In the main analysis and after PSM, adjuvant treatment was administered more frequently after MIPD. Croome et al. reported that a significantly higher proportion of patients undergoing OPD received delayed adjuvant treatment or did not receive adjuvant treatment; they also reported that LPD has certain advantages over OPD, such as shorter hospital stays and faster recovery times, allowing patients to recover and pursue adjuvant treatment options sooner [12]. Peng et al. reported that LPD patients had much shorter intervals between surgery and postoperative adjuvant chemotherapy than OPD patients [38]. Several studies have also found that minimally invasive surgery is associated with earlier initiation of chemotherapy, increased compliance, and improved survival rates for patients with colorectal cancer [39,40,41]. If laparoscopic surgery has these positive effects in other cancer patients, it may also improve outcomes for patients with PDAC. However, additional studies are required to evaluate the specific effects of MIPD on chemotherapy compliance among PDAC patients in the future.

Although there was no statistically significant difference, the lower mean number of harvested lymph nodes during MIPD compared with OPD may invite criticism that MIPD is not suitable for oncological pancreatic surgery, even when assuming that the extent of MIPD is similar to OPD. This result is similar to the result about PSM study of minimal invasive distal pancreatectomy for PDAC that the number of harvest LN and positive LN is less than that of open distal pancreatectomy [13]. However, Tomlinson et al. [42] reported that examination of 15 lymph nodes is optimal for the accurate staging of PDAC after PD. There have been several reports that extended lymphadenectomy does not yield significant survival benefits compared with standard resection in pancreatic head cancer [43,44,45]. In the current study, the mean number of harvested lymph nodes after MIPD was > 18. In addition, the average lymph node ratio and resection margin did not differ between the MIPD and OPD groups, and lymph node harvest showed no statistically significant difference compared with OPD in most studies published on LPD for PDAC [12,14,29,46]. The current study therefore suggests that lymph node harvest in itself is not oncologically problematic when performing MIPD for PDAC.

OS and DFS were not affected by the surgical approach in the current study. Chen et al. have recently produced a meta-analysis of studies of LPD for PDAC; the oncologic outcomes of LPD were seen to be equivalent to that of OPD, and LPD appeared promising in terms of long-term survival [47]. Croome et al. also reported longer DFS for LPD than OPD for PDAC [12]. There are, however, a number of reasons why these studies may have presented better survival outcomes relating to MIPD. First, the selection bias for implementing LPD was not considered even though postoperative pathologic outcomes were not statistically different. There was a trend toward smaller tumor size in the LPD group, although it did not reach a statistically significant difference in the meta-analysis [47]. This suggests that there is a tendency to perform MIPD in cases of less invasive PDAC. The current study implemented PSM to correct this limitation. Second, OPD was performed by different surgeons from those undertaking LPD in some studies [12]. This may affect the pathologic outcomes that affect survival rate. By contrast, the current study design meant that the same surgeons performed MIPD and OPD.

In the PSM analysis of the current study, the survival rate following MIPD was comparable to that of OPD. However, PSM analysis alone is not sufficient to confirm that there is no difference in survival between MIPD and OPD due to pathological differences between groups, such as perineural invasion. Differences in pathology were also found in PSM study of minimal invasive distal pancreatectomy for PDAC [13]. Because of these limitations, this study attempted to determine whether the surgical procedure affected survival regardless of pathology outcomes using a multivariate Cox proportional hazard model for patients selected by PSM. Comparative analysis of oncologic outcomes between the MIPD and OPD groups after PSM for pathologic outcome was performed, even though PSM analysis to include pathological results is statistically inadequate because the surgery itself may alter pathology. These results indicated that OS and DFS were not affected by the surgical procedure.

The current study has some limitations. Data were collected retrospectively, and the number of MIPD cases is low compared with OPD. Inherent selection bias may have occurred with patients slated for MIPD, as these subjects are more likely to be judged by the surgeon as having favorable outcomes, even though data was PSM-corrected. On this basis, the current results cannot be considered to be highly reliable. Therefore, randomized clinical trials are required to clarify the oncologic outcomes and survival rate associated with MIPD in the treatment of PDAC. Nevertheless, to the best of our knowledge, this study is the first to report a large-scale PSM analysis of MIPD for PDAC, and the results can therefore be considered to be meaningful.

## 5. Conclusions

This study of patients with PDAC demonstrates that MIPD resulted in shorter postoperative hospital stays and comparable perioperative and oncologic outcomes to OPD. Although future randomized studies will be required to validate these findings, MIPD can be considered as a safe, oncologically appropriate approach to the treatment of PDAC, and may be cautiously considered in selected PDAC patients.

## Figures and Tables

**Figure 1 cancers-12-00982-f001:**
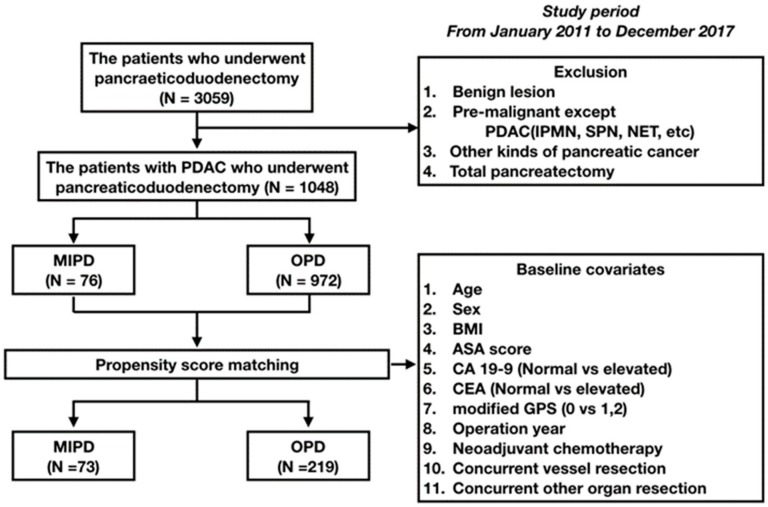
Patient flow diagram. We conducted a retrospective review of 1048 patients who underwent pancreatoduodenectomy after excluding 2011 patients based on the criteria listed in the figure. Of these, 76 patients underwent minimally invasive pancreatoduodenectomy and 972 patients underwent open pancreatoduodenectomy. After propensity score matching, 73 MIPD patients and 219 OPD patients were compared.

**Figure 2 cancers-12-00982-f002:**
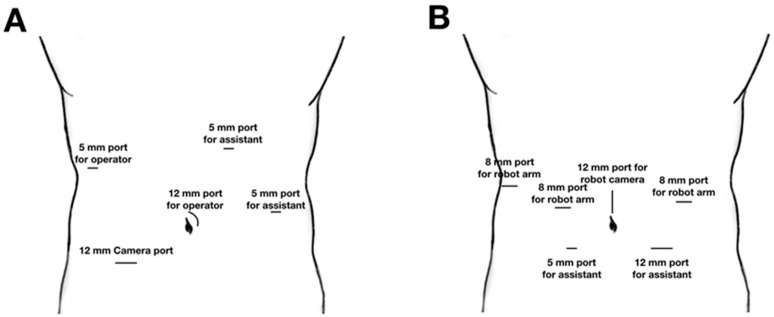
Trocar locations during MIPD. (**A**) Port locations for LPD and RPD performed during laparoscopic resection and anastomosis by robot instrument. (**B**) Port locations for RPD performed by robotic resection and anastomosis.

**Figure 3 cancers-12-00982-f003:**
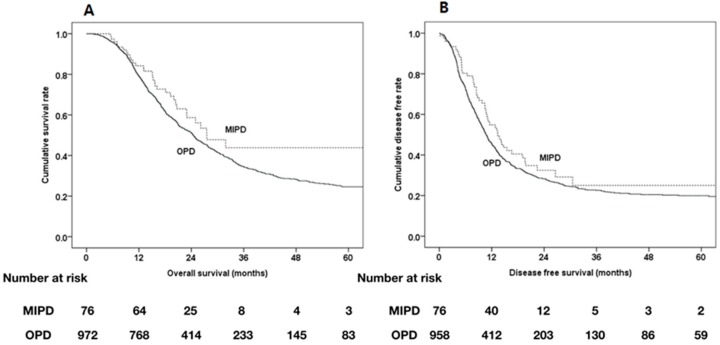
Kaplan-Meier survival curves of MIPD group (*n* = 76) and OPD group (*n* = 972). (A) The median OS was 27.6 months, and estimated 1-, 2-, 5- year OS were 84.2%, 58.7%, and 43.8%, respectively in the MIPD group, and 24.5 months and 79.3%, 51.2%, and 24.6% respectively, in the OPD group. Log rank *p*-value for this result was 0.079. (B) The median DFS was 13.4 months, and estimated 1-, 2-,5- year DFS were 54.8%, 32.5%, and 25.0%, respectively in the MIPD group, and 10.7 months and 45.6%, 28.3%, and 19.9%, respectively in the OPD group. Log rank *p*-value for this result was 0.151.

**Figure 4 cancers-12-00982-f004:**
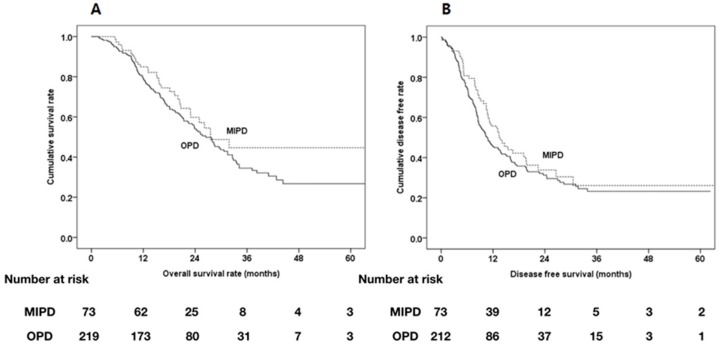
Kaplan-Meier survival curves of MIPD group (*n* = 73) and OPD group (*n* = 219) after propensity score matching. (A) Median OS and estimated 1-, 2-, and 5-year OS were 27.6 months and 84.9%, 59.8%, and 44.7% in the MIPD group, and 26.5 months, 79.4%, 54.0%, and 26.7% in the OPD group, respectively (*p* = 0.143). (B) Median DFS and estimated 1-, 2-, and 5-year DFS were 13.7 months and 55.7%, 33.8%, and 26.1% in the MIPD group and 10.7 months and 45.5%, 31.3%, and 23.2% in the ODP group, respectively (*p* = 0.278).

**Figure 5 cancers-12-00982-f005:**
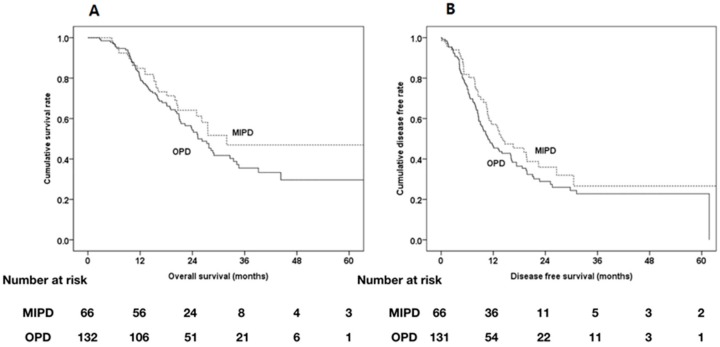
Kaplan-Meier survival curves of MIPD group (*n* = 66) and OPD group (*n* = 132) after propensity score matching with pathologic finding. (**A**) Median OS and estimated 1-, 2-, and 5-year OS were 31.9 months and 84.8%, 64.1%, and 47.0% in the MIPD group, and 25.3 months, 80.3%, 54.4%, and 29.6% in the OPD group, respectively (*p* = 0.177). (**B**) Median DFS and estimated 1-, 2-, and 5-year DFS were 14.1 months and 57.1%, 36.0%, and 26.6% in the MIPD group and 10.8 months and 46.3%, 28.9%, and 22.8% in the ODP group, respectively (*p* = 0.166).

**Table 1 cancers-12-00982-t001:** Patient demographics.

Variables		MIPD (*n* = 76)	OPD (*n* = 972)	*P*-Value
Mean age, years (±SD)		62.2 (±10.4)	61.9 (±9.8)	0.836
Sex, *n* (%)	Female	34 (44.7)	362 (37.2)	0.153
	Male	42 (55.3)	610 (62.8)	
Mean BMI, kg/m^2^ (±SD)		22.7 (±2.8)	22.8 (±2.94)	0.832
ASA score, *n* (%)	I	8 (10.5)	59 (6.1)	0.059
	II	60 (78.9)	858 (88.3)	
	III	8 (10.5)	55 (5.7)	
CA19-9, *n* (%)	Normal	34 (44.7)	311 (32.0)	0.017
	Increased	38 (50.0)	639 (65.7)	
	NA	4 (5.3)	22 (2.3)	
CEA, *n* (%)	Normal	53 (69.7)	747 (76.9)	> 0.999
	Increased	13 (17.1)	178 (18.3)	
	NA	10 (13.2)	47 (4.8)	
Preoperative biliary drainage, *n* (%)	Yes	36 (47.4)	603 (62.0)	0.016
	No	40 (52.6)	369 (38.0)	
mGPS, *n* (%)	0	56 (73.7)	700 (72.0)	0.459
	1–2	17 (22.4)	164 (16.9)	
	NA	3 (3.9)	108 (11.1)	
Neoadjuvant, *n* (%)	Yes	6 (7.9)	90 (9.3)	0.849
	No	70 (92.1)	882 (90.7)	
Concurrent vessel resection, n (%)	Vein	11 (14.5)	303 (31.2)	< 0.001
	Artery	1 (1.3)	35 (3.6)	
	Artery and vein	0 (0)	21 (2.2)	
	No	64 (84.2)	613 (63.1)	
Concurrent resection of another organ, *n* (%)	Yes	0 (0)	29 (3.0)	0.244
	No	76 (100)	943 (97.0)	
Year of surgery, *n* (%)	< 2015	10 (13.2)	520 (53.5)	< 0.001
	≥ 2015	66 (86.8)	452 (46.5)	

ASA, American Society of Anesthesiologists; BMI, body mass index; ENBD, endoscopic nasobiliary drainage; ERBD, endoscopic retrograde biliary drainage; mGPS, modified Glasgow prognostic score; MIPD, minimally invasive pancreatoduodenectomy; NA, not available; OPD, open pancreato-duodenectomy; PTBD, percutaneous transhepatic biliary drainage; SD, standard deviation.

**Table 2 cancers-12-00982-t002:** Perioperative outcome according to method of pancreaticoduodenectomy.

Variables		MIPD (*n* = 76)	OPD (*n* = 972)	*p*-Value
Mean operation time, minutes (±SD)		392 (±96)	368 (±99)	0.043
Overall complications, *n* (%)		23 (30.3)	349 (35.9)	0.322
In-hospital complications grade ^+^, n (%)	No	54 (71.1)	659 (67.8)	0.832
	Grade I–II	18 (23.7)	275 (28.3)	
	Grade III–V	4 (5.3)	38 (3.9)	
Late complications grade ^+^, *n* (%)	No	69 (90.8)	925 (95.2)	0.021
	Grade I–II	1 (1.3)	26 (2.7)	
	Grade III–V	6 (7.9)	21 (2.2)	
POPF ^++^, *n* (%) ^+^	No or	75 (98.7)	931 (95.8)	0.176
	biochemical leakage			
	Grade B or C	1 (1.3)	41 (4.2)	
Delayed gastric emptying B or C, *n* (%) *	Yes	2 (2.6)	30 (3.1)	>0.999
	No	74 (97.4)	942 (96.9)	
Post-pancreatectomy hemorrhage grade B or C, *n* (%) **	Yes	2 (2.6)	11 (1.1)	0.242
	No	74 (97.4)	961 (98.9)	
Biliary stricture during follow-up periods, *n*, (%)	Yes	4 (5.3)	7 (0.7)	0.006
	No	72 (94.7)	965 (99.3)	
Reoperation, *n*, (%)	Yes	2 (2.6)	18 (1.9)	0.651
	No	74 (97.4)	954 (98.1)	
90 day mortality, *n* (%)	Yes	0 (0)	7 (0.7)	0.589
	No	76 (100)	967 (99.3)	
Hospital stay after operation, days (±SD)	Mean	12.2 (±5.5)	15.0 (±8.6)	<0.001
Adjuvant	No	15 (19.7)	310 (31.9)	0.001
	CTx	44 (57.9)	499 (51.3)	
	CCRTx	14 (18.4)	162 (16.7)	
	RTx	3 (3.9)	1 (0.1)	
Adjuvant regimen	Fluorpyrimidine	17 (22.4)	303 (31.2)	0.113
	Gemcitabine based	41 (53.9)	270 (27.8)	
	FOLFIRINOX	0 (0.0)	11 (1.1)	
	NA	18 (23.7)	388 (39.9)	
Interval between surgery and adjuvant treatment, days (±SD)	Mean	47.0 (±16.2)	47.4 (±17.6)	0.860

^+^ Complication grade was classified according to the Clavien-Dindo classification. ^++^ POPF was graded according to the definition by the International Study Group of Pancreatic Surgery (ISGPS), updated in 2016. ^*^ Delayed gastric emptying defined in accordance with ISGPS specifications. ^**^ Post-pancreatectomy hemorrhage defined in accordance with ISGPS specifications. CTx, chemotherapy; CCRTx, concurrent chemoradiation therapy; FOLFIRINOX (fluorouracil, leucovorin, irinotecan, and oxaliplatin); MIPD, minimally invasive pancreatoduodenectomy; NA, not available; OPD, open pancreatoduodenectomy; POPF, postoperative pancreatic fistula; RTx, radiation therapy.

**Table 3 cancers-12-00982-t003:** Pathologic outcome according to surgical method of pancreaticoduodenectomy.

Variables		MIPD (*n* = 76)	OPD (*n* = 972)	*p*-Value
Mean pathologic tumor size, cm (±SD)		2.7 (±0.8)	3.1 (±1.0)	0.019
T stage (AJCC 8th), *n* (%)	T1	17 (22.4)	137 (14.1)	0.011
	T2	55 (72.4)	706 (72.6)	
	T3	4 (5.3)	122 (12.6)	
	T4	0 (0)	7 (0.7)	
N stage (AJCC 8th), *n* (%)	N0	30 (39.5)	373 (38.4)	0.330
	N1	37 (48.7)	411 (42.3)	
	N2	9 (11.8)	188 (19.3)	
Staging (AJCC 8th), ^+^ *n* (%)	IA	11 (14.5)	85 (8.7)	0.330
	IB	16 (21.1)	257 (26.4)	
	IIA	3 (3.9)	31 (3.2)	
	IIB	36 (47.4)	395 (40.6)	
	III	9 (11.8)	182 (18.7)	
	IV	1 (1.3)	22 (2.3)	
Differentiation	WD	6 (7.9)	113 (11.6)	0.798
	MD	59 (77.6)	721 (74.2)	
	PD	7 (9.2)	106 (10.9)	
	NA	4 (5.3)	32 (3.3)	
Lymphovascular invasion, *n* (%)	Yes	50 (65.8)	608 (62.6)	0.574
	No	26 (34.2)	364 (37.4)	
Perineural invasion, *n* (%)	Yes	53 (69.7)	852 (87.7)	<0.001
	No	23 (30.3)	120 (12.3)	
Mean number of harvested lymph nodes, *n* (±SD)		18.6 (±9.9)	22.1 (±10.6)	0.006
Mean number of positive lymph nodes, *n* (±SD)		1.5 (±1.9)	2.0 (±2.8)	0.041
Mean positive lymph node ratio, %, (±SD)		9.8 (±15.1)	9.8 (±13.4)	0.976
Resection margin ^++^, *n* (%)	R0	57 (75.0)	696 (71.6)	0.526
	R1	19 (25.0)	276 (28.4)	

^+^ TNM stage was graded according to American Joint Committee on Cancer Stage, 8th edition. ^++^ If closest safe resection margin was < 1 mm, it was categorized as R1. MD, moderately differentiated; MIPD, minimally invasive pancreatoduodenectomy; NA, not available; OPD, open pancreatoduodenectomy; PD, poorly differentiated; WD, well differentiated.

**Table 4 cancers-12-00982-t004:** Demographics of MIPD and OPD after propensity score matching.

Variables		MIPD (*n* = 73)	OPD (*n* = 219)	SMD
Mean age, years		62.4	63.3	0.091
Sex, *n* (%)	Female	32 (43.8)	105 (47.9)	0.083
	Male	41 (56.2)	114 (52.1)	
Mean BMI, kg/m^2^		22.79	22.85	0.021
ASA score, *n* (%)	I	7 (9.6)	23 (10.5)	0.054
	II	59 (80.8)	178 (81.3)	
	III	7 (9.6)	18 (8.2)	
CA19-9, *n* (%)	Normal	33 (45.2)	102 (46.6)	0.010
	Increased	36 (49.3)	109 (49.8)	
	NA	4 (5.5)	8 (3.7)	
CEA, *n* (%)	Normal	50 (68.5)	166 (75.8)	0.040
	Increased	13 (17.8)	39 (17.8)	
	NA	10 (13.7)	14 (6.4)	
Preoperative biliary drainage, *n* (%)	Yes	35 (47.9)	109 (49.8)	0.037
	No	38 (52.1)	110 (50.2)	
mGPS, *n* (%)	0	55 (75.3)	165 (75.3)	0.009
	1–2	15 (20.5)	44 (20.1)	
	NA	3 (4.1)	10 (4.6)	
Neoadjuvant, *n* (%)	Yes	6 (8.2)	15 (6.8)	0.052
	No	67 (91.8)	204 (93.2)	
Concurrent vessel resection, *n* (%)	Yes	12 (16.4)	39 (17.8)	0.036
	No	61 (83.6)	180 (82.2)	
Concurrent resection of another organ, *n* (%)	Yes	0 (0)	0 (0)	<0.001
	No	73 (100)	219 (100)	
Year of surgery, *n* (%)	<2015	10 (13.7)	31 (14.2)	0.013
	≥2015	63 (86.3)	188 (85.8)	

BMI, body mass index; mGPS, modified Glasgow prognostic score; MIPD, minimally invasive pancreatoduodenectomy; NA, not available; OPD, open pancreatoduodenectomy; SMD, standardized mean difference.

**Table 5 cancers-12-00982-t005:** Perioperative outcome of MIPD and OPD after propensity score matching.

Variables		MIPD (*n* = 73)	OPD (*n* = 219)	*p*-Value ^#^
Mean operation time, minutes		392	327	<0.001
Overall complications, *n*, (%)		23 (31.5)	91 (41.6)	0.128
In-hospital complications grade ^+^, *n* (%)	No	51 (69.9)	142 (64.8)	0.781
	Grade I–II	18 (23.7)	68 (31.1)	
	Grade III–V	4 (5.3)	9 (4.1)	
Late complications grade ^+^, *n* (%)	No	66 (90.4)	202 (92.2)	0.202
	Grade I–II	1 (1.4)	10 (4.6)	
	Grade III–V	6 (8.2)	7 (3.2)	
POPF ^++^, *n* (%)	No or Biochemical leakage	72 (98.6)	214 (97.7)	0.640
	Grade B–C	1 (1.4)		
Delayed gastric emptying B or C, *n* (%) *	Yes	2 (2.7)	6 (2.7)	>0.999
Post-pancreatectomy hemorrhage grade B or C, *n* (%) **	Yes	2 (2.7)	1 (0.5)	0.140
Biliary stricture during follow-up periods, *n*, (%)	Yes	4 (5.5)	4 (1.8)	0.110
Reoperation, *n* (%)	Yes	2 (2.7)	4 (1.8)	0.630
Mean hospital stay after surgery, days		12.4	14.2	0.040
Adjuvant treatment	Yes	59 (80.8)	128 (59.8)	0.002
Adjuvant regimen	Fluoropyrimidine	16 (21.9)	47 (21.5)	0.191
	Gemcitabine based	40 (54.8)	74 (33.8)	
	or FOLFIRINOX			
	NA	17 (23.3)	98 (44.7)	
Mean interval between surgery and adjuvant treatment (±SD)		47.3 (±16.1)	46.6(±15.8)	0.740

^+^ Complication grade was classified according to the Clavien-Dindo classification.^++^ POPF was graded according to the definition updated in 2016 by the International Study Group Pancreatic Fistula. * Delayed gastric emptying defined in accordance with ISGPS specifications. ** Post-pancreatectomy hemorrhage defined in accordance with ISGPS specifications. ^#^ Calculated using a generalized estimating equation with exchangeable correlation structure within matched stratum. FOLFIRINOX, fluorouracil, leucovorin, irinotecan, and oxaliplatin; MIPD, minimally invasive pancreatoduodenectomy; OPD, open pancreatoduodenectomy; POPF, postoperative pancreatic fistula.

**Table 6 cancers-12-00982-t006:** Pathologic outcome of MIPD and OPD after propensity score matching.

Variables		MIPD (*n* = 73)	OPD (*n* = 219)	*p*-Value *
Mean pathologic tumor size, cm		2.75	2.84	0.49
T stage (AJCC 8th), *n* (%)	T1	15 (20.5)	37 (16.9)	0.415
	T2	54 (74.0)	166 (75.8)	
	T3	4 (5.5)	16 (7.3)	
	T4	0 (0.0)	0 (0.0)	
N stage (AJCC 8th), *n* (%)	N0	29 (39.7)	103 (47.0)	0.573
	N1	35 (47.9)	76 (34.7)	
	N2	9 (12.3)	40 (18.3)	
Staging (AJCC 8th) ^+^, *n* (%)	IA	10 (13.7)	28 (12.8)	0.444
	IB	16 (21.9)	70 (32.0)	
	IIA	3 (4.1)	6 (2.7)	
	IIB	34 (46.6)	74 (33.8)	
	III	9 (12.3)	36 (16.4)	
	IV	1 (1.4)	5 (2.3)	
Differentiation	WD	6 (8.2)	28 (12.8)	0.286
	MD	58 (79.5)	168 (76.7)	
	PD	6 (8.2)	16 (7.3)	
	NA	3 (4.1)	7 (3.2)	
Lymphovascular invasion, *n* (%)	Yes	48 (65.8)	127 (58.0)	0.243
	No	25 (34.2)	92 (42.0)	
Perineural invasion, *n* (%)	Yes	51 (69.9)	179 (81.7)	0.042
	No	22 (30.1)	40 (18.3)	
Mean number of harvested lymph nodes, *n*		18.9	21.3	0.073
Mean number of positive lymph nodes, *n*		1.52	1.94	0.82
Mean positive lymph node ratio, %		9.52	9.17	0.86
Resection margin ^++^, *n* (%)	R0	56 (76.7)	164 (74.9)	0.76
	R1	17 (23.3)	55 (25.1)	

^+^ TNM stage was graded according to American Joint Committee on Cancer Stage, 8th edition. ^++^ If closest safe resection margin was < 1 mm, it was categorized as R1. * Calculated using a generalized estimating equation (GEE) with exchangeable correlation structure within matched stratum. For ordered outcome, Heagerty and Zeger [1996, Journal of the American Statistical Society 91, 1024-1036] GEE approach was used for clustered ordinal measurements. MD, moderately differentiated; MIPD, minimally invasive pancreatoduodenectomy; NA, not available; OPD, open pancreatoduodenectomy; PD, poorly differentiated; WD, well differentiated.

**Table 7 cancers-12-00982-t007:** Multivariable model of risk factors for overall survival after propensity score matching.

Variables		HR	95% CI	*p*-Value
Type of surgery	OPD	Ref	-	0.24
	MIPD	0.786	0.525–1.175	
Size		1.092	0.847–1.409	0.496
T stage (AJCC 8th)	T1	Ref	-	0.452
	T2	1.397	0.776–2.514	0.265
	T3	1.881	0.662–5.341	0.236
N stage (AJCC 8th)	N0	Ref	-	0.125
	N1	1.501	0.972–2.317	0.067
	N2	1.958	0.980–3.913	0.057
M stage (AJCC 8th)	M1	1.229	0.682–2.214	0.493
Differentiation	WD	Ref	-	<0.001
	MD	1.826	1.085–3.074	
	PD	4.962	2.664–9.245	
Lymphovascular invasion	No	Ref	-	0.122
	Yes	1.411	0.912–2.184	
Perineural invasion	No	Ref	-	0.026
	Yes	1.833	1.074–3.128	
Number of harvested lymph nodes		1	0.978–1.022	0.991
Number of positive lymph nodes		1.046	0.924–1.183	0.478
Positive lymph node ratio		0.994	0.966–1.022	0.657
Resection margin	R0	Ref	-	0.001
	R1	1.793	1.268–2.536	

CI, confidence interval; HR, hazard ratio; MD, moderately differentiated; NA, not available; PD, poorly differentiated; WD, well differentiated.

**Table 8 cancers-12-00982-t008:** Multivariable model of risk factors for disease-free survival after propensity score matching.

Variables		HR	95% CI	*p*-Value
Type of surgery	OPD	Ref	-	0.206
	MIPD	0.776	0.523–1.150	
Size		1.194	0.957–1.489	0.116
T stage (AJCC 8th)	T1	Ref	-	0.909
	T2	0.921	0.557–1.523	0.748
	T3	0.793	0.281–2.242	0.662
N stage (AJCC 8th)	N0	Ref	-	0.075
	N1	1.674	1.061–2.641	0.027
	N2	1.901	0.706–5.116	0.204
M stage (AJCC 8th)	M1	1.209	0.420–3.482	0.725
Differentiation	WD	Ref	-	<0.001
	MD	1.357	0.911–2.023	0.133
	PD	3.293	1.838–5.899	<0.001
Lymphovascular invasion	No	Ref	-	0.21
	Yes	1.299	0.863–1.954	
Perineural invasion	No	Ref	-	0.067
	Yes	1.487	0.973–2.272	
Number of harvested lymph nodes		1.002	0.984–1.020	0.853
Number of positive lymph nodes		1.044	0.860–1.267	0.661
Positive lymph node ratio		0.994	0.972–1.016	0.595
Resection margin	R0	Ref	-	0.211
	R1	1.255	0.879–1.793	

CI, confidence interval; HR, hazard ratio; MD, moderately differentiated; NA, not available; PD, poorly differentiated; WD, well differentiated.

**Table 9 cancers-12-00982-t009:** Demographics and pathologic outcome of MIPD and OPD after propensity score matching with pathologic finding.

Variables		MIPD (*n* = 66)	OPD (*n* = 132)	SMD
Mean age, years		62.1	62.5	0.035
Sex, *n* (%)	Female	29 (43.9)	57 (43.2)	0.015
Mean BMI, kg/m^2^		22.91	22.83	0.028
ASA score, *n* (%)	I	5 (7.6)	10 (7.6)	0.027
	II	55 (83.3)	111 (84.1)	
	III	6 (9.1)	11 (8.3)	
CA19-9, *n* (%)	Increased	33 (50.0)	67 (50.1)	0.018
CEA, *n* (%)	Increased	12 (18.2)	25 (18.9)	0.043
Preoperative biliary drainage, *n* (%)	Yes	33 (50.0)	60 (45.5)	0.091
mGPS, *n* (%)	1–2	12 (18.2)	28 (21.2)	0.087
Neoadjuvant, *n* (%)	Yes	6 (9.1)	10 (7.6)	0.055
Concurrent vessel resection, *n* (%)	Yes	12 (18.2)	25 (18.9)	0.019
Concurrent resection of another organ, *n* (%)	Yes	0 (0)	0 (0)	<0.001
Year of surgery, *n* (%)	≥2015	57 (86.4)	111 (84.1)	0.064
Mean pathologic tumor size, cm	Mean	2.78	2.77	0.01
Staging (AJCC 8^th^)^+^, *n* (%)	IA	9 (13.6)	20 (15.2)	0.15
	IB	16 (24.2)	30 (22.7)	
	IIA	3 (4.5)	5 (3.8)	
	IIB	28 (42.4)	61 (46.2)	
	III	9 (13.6)	13 (9.8)	
	IV	1 (1.5)	3 (2.3)	
Differentiation	WD	6 (9.1)	15 (11.4)	0.158
	MD	54 (81.8)	101 (76.5)	
	PD	3 (4.5)	10 (7.6)	
	NA	3 (4.5)	6 (4.5)	
Lymphovascular invasion, *n* (%)	Yes	41 (62.1)	83 (62.9)	0.016
Perineural invasion, *n* (%)	Yes	49 (74.2)	93 (70.5)	0.085
Mean number of harvested lymph nodes, *n*		19.4	20	0.066
Mean number of positive lymph nodes, *n*		1.53	1.55	0.007
Mean positive lymph node ratio, %		8.29	8.46	0.016
Resection margin^++^, *n* (%)	R1	16 (24.2)	35 (26.5)	0.052

^+^ TNM stage was graded according to American Joint Committee on Cancer Stage, 8th edition. ^++^ If closest safe resection margin was < 1 mm, it was categorized as R1. BMI, body mass index; mGPS, modified Glasgow prognostic score; NA, not available; SMD, standardized mean difference.

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
