# Peer review of "Comparison of Minimally Invasive versus Open Pancreatoduodenectomy for Pancreatic Ductal Adenocarcinoma: A Propensity Score Matching Analysis"

_cancers, 2020, doi:10.3390/cancers12040982_

Round 1

Reviewer 1 Report

In this manuscript, authors present a study where they compared between minimally invasive surgery and open surgery on treating pancreatic ductal adenocarcinoma using a number of metrics after viewed retrospectively 1048 patients. The authors have done adequate analysis to compare the two surgical techniques that are used in a particular cancer type side by side, which is a pilot study in this field. The information from this study would be of great interest to the audients of this journal.

The work from a group of surgeons across Europe has published a study addressing the same issue: a comparison between key hole surgery and open surgery for PDAC (van Hilst, J., de Rooij, T., Klompmaker, S., Rawashdeh, M., Aleotti, F., Al-Sarireh, B., … (E-MIPS), for the E. C. on M. I. P. S. (2019). Minimally Invasive versus Open Distal Pancreatectomy for Ductal Adenocarcinoma (DIPLOMA): A Pan-European Propensity Score Matched Study. Annals of Surgery, 269(1) ). Please include this relevant study and discuss if the main findings are match or disagree.

Other comments are noted in the pdf file.

Author Response

Reviewer #1: In this manuscript, authors present a study where they compared between minimally invasive surgery and open surgery on treating pancreatic ductal adenocarcinoma using a number of metrics after viewed retrospectively 1048 patients. The authors have done adequate analysis to compare the two surgical techniques that are used in a particular cancer type side by side, which is a pilot study in this field. The information from this study would be of great interest to the audients of this journal.

The work from a group of surgeons across Europe has published a study addressing the same issue: a comparison between key hole surgery and open surgery for PDAC (van Hilst, J., de Rooij, T., Klompmaker, S., Rawashdeh, M., Aleotti, F., Al-Sarireh, B., … (E-MIPS), for the E. C. on M. I. P. S. (2019). Minimally Invasive versus Open Distal Pancreatectomy for Ductal Adenocarcinoma (DIPLOMA): A Pan-European Propensity Score Matched Study. Annals of Surgery, 269(1) ). Please include this relevant study and discuss if the main findings are match or disagree.

Other comments are noted in the pdf file.

Response : We thank the reviewer for this comment. As you recommended, we mentioned the article in the introduction and discussion section.

(1st paragraph, introduction section)

Minimally invasive surgery has become the standard of care for many surgical procedures across different specialties and is currently standard procedure for the resection of intraabdominal organs, including the stomach [1,2], gallbladder [3], spleen [4,5], colon [6,7], and kidney [8,9]. However, its use in pancreatic surgery has been limited because of the complexity of these operations. Recently, minimally invasive surgery for benign and malignant pancreatic tumors has gained wider acceptance and is attracting more research attention [10-12]. In addition, a Pan-European propensity score matched study was published that showed a comparable survival outcome when performing minimal invasive distal pancreatectomy for pancreatic cancer [13]. However, minimally invasive pancreatoduodenectomy (MIPD), which includes laparoscopic pancreatoduodenectomy (LPD) and robotic pancreatoduodenectomy (RPD), remains limited by a lack of generalizability, and open surgery is preferred for pancreatic ductal adenocarcinoma (PDAC) due to concerns about adequate oncological outcomes and the potential for vessel resection. Recent studies have investigated LPD for PDAC [12,14]. Until now, however, no matched analyses have been conducted to analyze the perioperative and long-term oncologic outcomes of MIPD versus open pancreatoduodenectomy (OPD) for PDAC. The present retrospective study has compared perioperative and oncological outcomes between MIPD and OPD for PDAC using propensity score matching (PSM) analysis.

(5th paragraph, discussion section)

Although there was no statistically significant difference, the lower mean number of harvested lymph nodes during MIPD compared with OPD may invite criticism that MIPD is not suitable for oncological pancreatic surgery, even when assuming that the extent of MIPD is similar to OPD. This result is similar to the result about PSM study of minimal invasive distal pancreatectomy for PDAC that the number of harvest LN and positive LN is less than that of open distal pancreatectomy [13]. However, Tomlinson et al.[42] reported that examination of 15 lymph nodes is optimal for the accurate staging of PDAC after PD. There have been several reports that extended lymphadenectomy does not yield significant survival benefits compared with standard resection in pancreatic head cancer [43-45]. In the current study, the mean number of harvested lymph nodes after MIPD was > 18. In addition, the average lymph node ratio and resection margin did not differ between the MIPD and OPD groups, and lymph node harvest showed no statistically significant difference compared with OPD in most studies published on LPD for PDAC [12,14,29,46]. The current study therefore suggests that lymph node harvest in itself is not oncologically problematic when performing MIPD for PDAC.

(7th paragraph, discussion section)

In the PSM analysis of the current study, the survival rate following MIPD was comparable to that of OPD. However, PSM analysis alone is not sufficient to confirm that there is no difference in survival between MIPD and OPD due to pathological differences between groups, such as perineural invasion. Differences in pathology were also found in PSM study of minimal invasive distal pancreatectomy for PDAC[13]. Because of these limitation, this study attempted to determine whether the surgical procedure affected survival regardless of pathology outcomes using a multivariate Cox proportional hazard model for patients selected by PSM. Comparative analysis of oncologic outcomes between the MIPD and OPD groups after PSM for pathologic outcome was performed, even though PSM analysis to include pathological results is statistically inadequate because the surgery itself may alter pathology. These results indicated that OS and DFS were not affected by the surgical procedure.

Reviewer 2 Report

This manuscript titled “Comparison of minimally invasive versus open pancreatoduodenectomy for pancreatic ductal adenocarcinoma: a propensity score matching analysis”, was presented by Jaewoo Kwon and coworkers, aimed to evaluated the difference between MIPD and OPD in PDAC. The authors showed basically no obvious difference except a shorter postoperative hospital stays. However, there still a comment needs to be addressed by the authors:

Line86 indicated recurrence was diagnosed based on detection of new progressive lesions on abdominal CT and an increase in CA19-9 levels. What's the difference of recurrence/metastasis rate between the 2 surgeries?  What’s the rate of disease-free survival? I ask this because there are 2 studies showing shortage of minimally invasive (N Engl J Med 379:1895-1904,2018. and N Engl J Med 379:1905-1914,2018.) I only point the minimally invasive surgery, but not the Cervical Cancer

The whole manuscript is well written and explanation is appropriate. The discussion about LPD is good, though it looks promising in terms of long-term survival.

Author Response

Reviewer #2: This manuscript titled “Comparison of minimally invasive versus open pancreatoduodenectomy for pancreatic ductal adenocarcinoma: a propensity score matching analysis”, was presented by Jaewoo Kwon and coworkers, aimed to evaluated the difference between MIPD and OPD in PDAC. The authors showed basically no obvious difference except a shorter postoperative hospital stays. However, there still a comment needs to be addressed by the authors:

Line86 indicated recurrence was diagnosed based on detection of new progressive lesions on abdominal CT and an increase in CA19-9 levels. What's the difference of recurrence/metastasis rate between the 2 surgeries?  What’s the rate of disease-free survival? I ask this because there are 2 studies showing shortage of minimally invasive (N Engl J Med 379:1895-1904,2018. and N Engl J Med 379:1905-1914,2018.) I only point the minimally invasive surgery, but not the Cervical Cancer

The whole manuscript is well written and explanation is appropriate. The discussion about LPD is good, though it looks promising in terms of long-term survival.

Response :  We thank the reviewer for this comment. We added 1-, 2- year DFS rate with confidence interval in result section. There was no difference in DFS between the two surgeries even at 1 or 2 years after surgery. Estimated 1-, 2- year DFS rate was 55.7% (95% CI, 45.4%–68.5%), 33.8% (95% CI, 23.8%–48.1%) in MIPD group and 45.5%(95% CI, 39.1%–53.0%), 31.3% (95% CI, 23.8%–48.1%) in OPD group. In addition, in this study, multivariable model of risk factors for disease-free survival after propensity score matching(table 8) was used to further confirm whether surgery affects DFS, and the result was the same (MIPD; Harzard ratio=0.776 ; CI, 0.523–1.150; p = 0.206). It is thought that more cases and randomized clinical trials are needed to conclude whether DFS is better in MIPD for PDAC.

(7th paragraph, result section)

The median OS and DFS were similar in the two groups (Figure 4). Estimated 1-, 2- year OS rate was 84.9% (95% CI, 77.1%–93.5%), 59.8% (95% CI, 48.3%–74.1%) in MIPD group and 79.4% (95% CI, 74.2%–84.9%), 54.0% (95% CI, 47.4%–61.6%) in OPD group. There was no difference in survival rate between two groups (p = 0.143). Estimated 1-, 2- year DFS rate was 55.7% (95% CI, 45.4%–68.5%), 33.8% (95% CI, 23.8%–48.1%) in MIPD groups and 45.5%(95% CI, 39.1%–53.0%), 31.3% (95% CI, 23.8%–48.1%) in OPD group. No notable differences were seen in DFS between two group (p = 0.278).